# Computationally predicting clinical drug combination efficacy with cancer cell line screens and independent drug action

Alexander Ling [1,2] & R. Stephanie Huang [1✉]

Evidence has recently emerged that many clinical cancer drug combinations may derive their efficacy from independent drug action (IDA), where patients only receive benefit from the single most effective drug in a drug combination. Here we present IDACombo, an IDA based method to predict the efficacy of drug combinations using monotherapy data from high-throughput cancer cell line screens. We show that IDACombo predictions closely agree with measured drug combination efficacies both in vitro (Pearson's correlation = 0.93 when comparing predicted efficacies to measured efficacies for >5000 combinations) and in a systematically selected set of clinical trials (accuracy > 84% for predicting statistically significant improvements in patient outcomes for 26 first line therapy trials). Finally, we demonstrate how IDACombo can be used to systematically prioritize combinations for development in specific cancer settings, providing a framework for quickly translating existing monotherapy cell line data into clinically meaningful predictions of drug combination efficacy.

[1] Department of Experimental and Clinical Pharmacology, University of Minnesota, Minneapolis, MN 55455, USA. [2] Committee on Cancer Biology, University of Chicago, Chicago, IL 60637, USA. ✉email: rshuang@umn.edu

Drug combinations are a cornerstone of modern therapy for many different cancers[1–3], but the vast number of possible drug combinations (many orders of magnitude greater than the number of possible monotherapies) makes it infeasible to screen them all experimentally when searching for new combination therapies. To overcome this problem, efforts have been made to develop computational methods that can identify promising drug combinations before experimentally testing them.

So far, these methods have mostly focused on estimating drug synergy, where the effect of a drug combination is greater than the additive effect of the drugs in the combination. Such models have been developed using a variety of approaches, including those based on mechanistic understandings, drug similarity, known interaction frequencies, and machine learning[4,5]. A recent collaborative effort to improve these models gave 160 research teams access to one of the largest available in vitro drug combination screens and tasked them with developing novel approaches for predicting drug synergy based on information such as gene expression, monotherapy response, drug structure, and drug mechanisms[6]. While many of the developed methods performed near the limits of experimental reproducibility with in-sample validation, out-of-sample validation against an independent screen[7] resulted in performance that was little better than random classification. Furthermore, a meta-analysis of 86 clinical articles suggests that current pre-clinical measurements of synergy are not associated with clinical trial results[8]. These results suggest that significant challenges remain to be overcome before methods of predicting drug synergy become clinically useful on a large scale.

Given the urgent need for computational models to predict drug combination efficacy and the challenges associated with models based on drug synergy, we chose to develop a model based on an assumption other than synergy. Since the earliest drug combination trials in cancer, researchers have considered the possibility that drug combinations confer patient benefit via drug independence rather than drug synergy[9]. While there are multiple theories for how drug combination efficacies should be calculated when drugs act independently[10], we chose to focus on independent drug action (IDA), which hypothesizes that the expected effect of a combination of non-interacting drugs is simply the effect of the single most effective drug in the combination. Evidence for the clinical relevance of this model was recently published[11]. Furthermore, the simplicity of IDA allows us to directly calculate drug combination efficacy using monotherapy drug screening data without the need for large training datasets with measured drug combination efficacies. Since numerous large in vitro monotherapy drug screening datasets already exist[12], this allows efficacy predictions for hundreds of thousands of two-drug combinations and hundreds of millions of three-drug and four-drug combinations to be made in a matter of weeks to months, whereas experimentally testing even a subset of those combinations could require decades. Such an approach holds enormous potential for helping researchers quickly identify drugs that can either be effectively combined with existing therapies or used in completely novel combination therapies.

Here we present the IDACombo method, which uses experimentally measured in vitro monotherapy response data to predict drug combination efficacies using the IDA model. We validate its predictions independently using both in vitro and patient clinical datasets. Furthermore, we prospectively predict the efficacies of thousands of two-drug combinations in 27 cancer types/subtypes and demonstrate how those predictions can be used to quickly identify candidate drug combinations for future clinical development.

## Results

**Design principle and workflow.** IDACombo relies on the principle of IDA, predicting that the efficacy of a drug combination in a given cell line or patient will be equal to the effect of the single best drug in that combination (Fig. 1a). Importantly, IDACombo predictions are concentration dependent, which allows us to predict combination efficacy specifically when each drug is used at its clinically relevant concentration. Furthermore, predictions represent an average response across populations of cell-lines/patients, which mimics the way treatment efficacies are measured in clinical trials.

We performed in vitro validation of IDACombo by directly comparing predicted combination viabilities to experimentally measured combination viabilities from published in vitro drug combination screens (Fig. 1b). Validation against published clinical trial results was more complicated, because the results of a clinical trial depend not only on the magnitude of the difference in efficacy between a test and control therapy, but also on the number of events (i.e. tumor progressions, patient deaths, etc.) observed by the trial. Thus, to make a meaningful comparison between IDACombo predictions and clinical trial results, we chose to treat the remaining percent viability after drug treatment as an estimate of hazard for patients treated with that therapy. This allowed us to estimate hazard ratios (HRs) for paired test and control therapies tested in clinical trials, which we then used to estimate statistical powers for each trial (see "Methods" section). Trials were classified as likely to succeed or fail using an 80% power threshold (Fig. 1c). Prospective analyses to identify novel efficacious drug combinations were performed either in a high-throughput fashion using summary statistics to compare many combinations at once or in a focused fashion where the efficacies of individual combinations were assessed using a range of drug concentrations (Fig. 1d).

**In-sample validation within in vitro drug combination datasets.** We first compared predictions made with IDACombo to measure combination efficacies for ~5000 drug combinations available in the NCI-ALMANAC dataset[13]. Monotherapy data from NCI-ALMANAC was used to predict mean viabilities for each drug combination in the dataset, and the predicted combination viabilities were compared to the measured combination viabilities (Fig. S1A). This revealed that IDACombo predictions in NCI-ALMANAC strongly correlate with measured combination viabilities (Pearson's $r = 0.932$, Spearman's rho $= 0.929$, Fig. 2a, Supplementary Data 1). Furthermore, the large majority of predicted efficacies were within 10% viability of the observed values with a median error of 3.22% viability (Fig. 2b), and the predictions were slightly skewed towards being conservative predictions, with 62.5% of combination efficacies being predicted to be less effective than they actually are and 37.5% predicted to be more effective than they actually are (Fig. 2c). Associations between predicted and measured combination efficacies in NCI-ALMANAC are plotted on a drug-by-drug basis in Supplementary Data 2 as a resource for researchers who are interested in seeing detailed results for their particular drugs of interest. To ensure that these results were not specific to the NCI-ALMANAC dataset, we repeated this analysis using two smaller drug combination datasets: the AstraZeneca–Sanger Drug Combination Prediction DREAM Challenge dataset[6] (AZ–S DREAM Challenge) and the drug combination dataset from O'Neil et al. (2016)[7]. IDACombo showed the same high correlation between predicted and measured drug combination efficacy and low prediction error in these datasets as in NCI-ALMANAC (Fig. S2).

While outliers are certainly present in all three datasets where predicted viabilities for a combination deviate significantly from

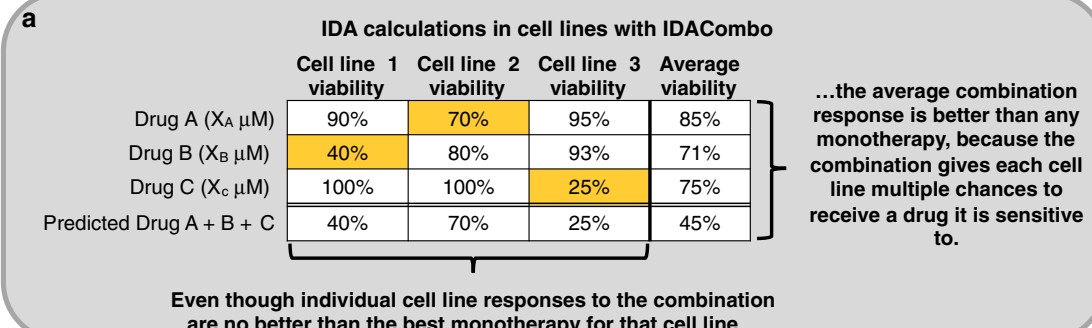

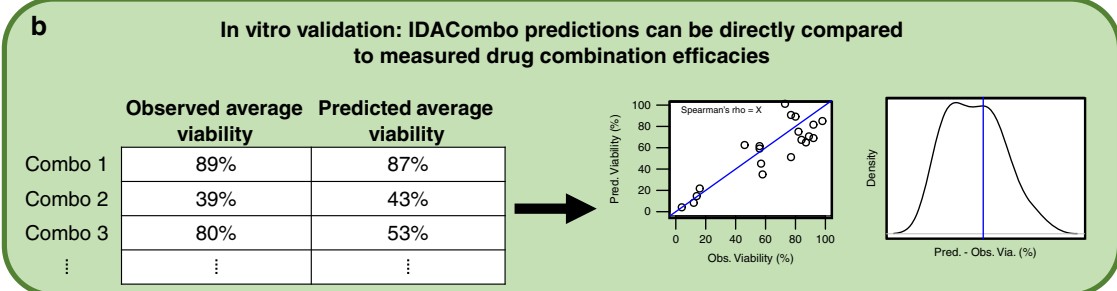

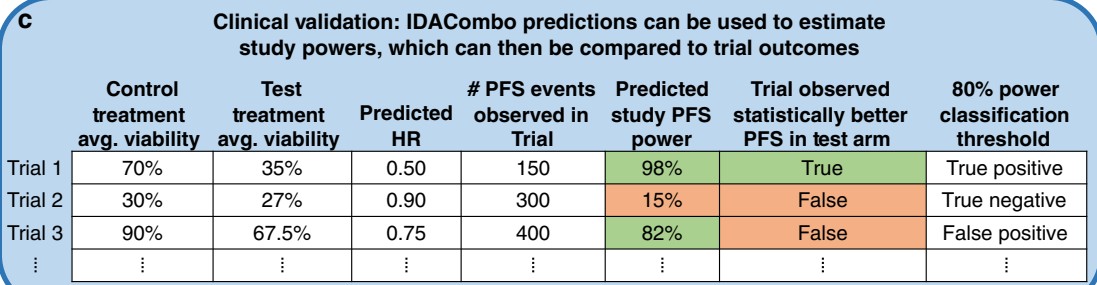

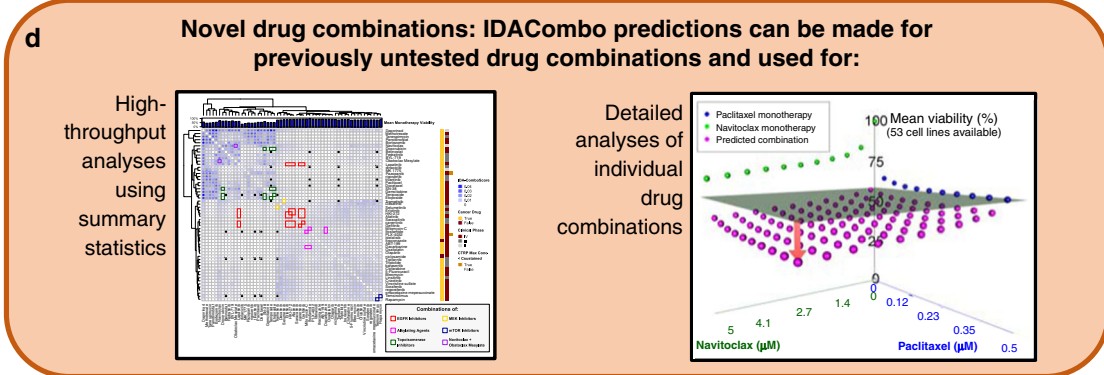

**Fig. 1 IDACombo allows drug combination efficacy predictions to be made using monotherapy cell line screening data, and these predictions can be validated against measured efficacies or used to identify novel efficacious drug combinations. a** Example calculations demonstrating how IDACombo predicts drug combination efficacies based on IDA. In this example, three cell lines (1–3) with measured efficacies for three monotherapies (A–C) at their selected concentrations are used to predict the efficacy of the combination of drugs A + B + C. Highlighted cells indicate the best monotherapy for that cell line (i.e. provides greatest reduction in viability). **b** Strategy for validating IDACombo efficacy predictions using in vitro measurements of efficacy. Measured and predicted average efficacies for each treatment can be directly compared by calculating their correlation and calculating prediction errors. **c** Strategy for validating IDACombo efficacy predictions using published clinical trial results. Combination efficacies which have been predicted using cell line data are used to predict study HRs, and these predicted HRs are used, along with the number of events observed in each clinical trial, to predict study powers. Predicted HRs can be compared to reported HRs, and a power threshold (80%) can be set to classify trials as likely or unlikely to detect a significant improvement in a trial outcome (i.e. PFS), which can then be compared to observed trial outcomes. **d** Analysis techniques available for using IDACombo predictions to identify novel efficacious drug combinations. High-throughput analyses using summary statistics can be used to compare efficacy predictions for many drug combinations at once, or detailed analyses can be used to explore the efficacy of a single drug combination at varying concentrations of each drug in the combination.

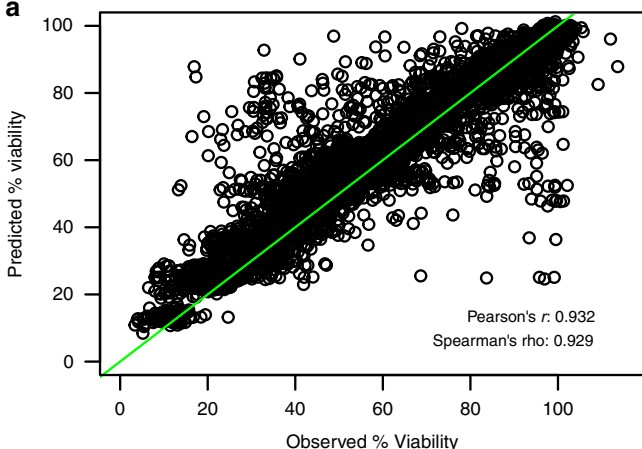

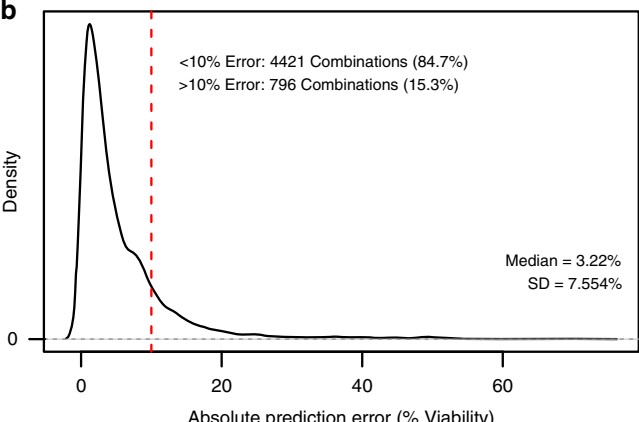

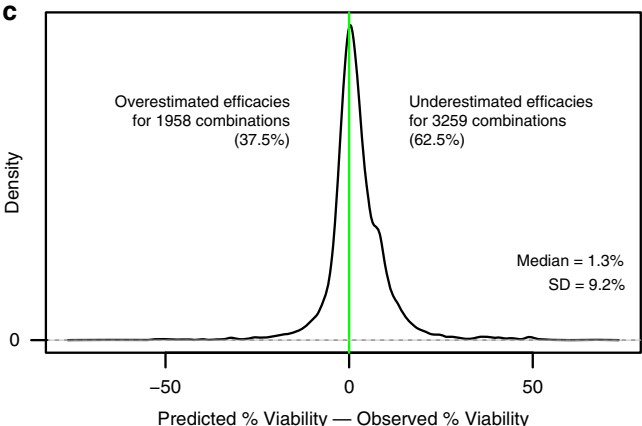

**Fig. 2 Agreement between predicted and observed combination viabilities in NCI-ALMANAC. a** Scatterplot showing high correlation between predicted average percent viability and experimentally observed average percent viability for each drug combination in NCI-ALMANAC. Predictions were made using monotherapy data from the dataset. The green line is a reference diagonal with slope = 1 and intercept = 0. Note that predictions were only made for the maximum concentration tested for each drug. **b** Density plot showing that the absolute values of the differences between the predicted percent viabilities and the observed percent viabilities for each drug combination are generally below 10%, with >50% of drug combinations having an absolute prediction error below 5%. The red line marks a difference of ±10% viability between predicted and observed values. **c** Density plot showing that the differences between the predicted percent viabilities and the observed percent viabilities for each drug combination have a slight tendency towards being positive—indicating that IDA-Combo underestimates efficacy more often than it overestimates efficacies. Source data are provided as a Source Data file.

(GDSC)[14] and the Cancer Therapeutics Response Portal Version 2 (CTRPv2)[15] monotherapy cell line screening datasets to make IDACombo predictions for all drug combinations in NCI-ALMANAC which used drugs that were also present in GDSC and CTRPv2. This approach resulted in weaker correlations between predicted and measured combination viabilities than the previous in-sample validations (Spearman's rho = 0.59 and 0.65 for CTRPv2 and GDSC, respectively, Fig. S3A and S3B). However, these correlations approach the limits of measured agreement between CTRPv2 and GDSC monotherapy viabilities with NCI-ALMANAC viabilities (Spearman's rho = 0.599 and 0.596 for CTRPv2 and GDSC, respectively, Figs. S3C and S3D). These results suggest that IDACombo predictions are robust across datasets within the limits of variances due to technical error, differences in screening methodologies, and differences in the size and composition of the screened cell line populations.

**Identifying clinical trials and drug concentrations for clinical validation of IDACombo.** While the in vitro validation results suggest that the IDACombo approach is promising, the true measure of the model's utility is its ability to accurately predict the efficacy of drug combinations in the clinical setting. To explore this, we sought to validate IDACombo predictions of clinical trial efficacy against published clinical trial results. The full pipeline for this clinical validation is outlined in Fig. S1B.

We first identified a diverse and unbiased set of clinical trials against which we could test IDACombo's predictions. Ultimately, this resulted in the identification of 54 usable clinical trials which tested 62 unique treatments involving 32 drugs (Fig. 3, Supplementary Data 3). Given the concentration-dependent nature of IDACombo's predictions, we also searched the published literature to identify clinically relevant concentrations for each drug at each dose used in these trials. This process is described in the Supplemental Text, Methods, and Fig. S4. The selected concentrations are included in Supplementary Data 4.

measured viabilities and drug synergy/additivity or antagonism may exist, these results suggest that most drug combinations in the NCI-ALMANAC, AZ-S DREAM Challenge, and O'Neil et al. datasets can be accurately modeled via IDA, and they support the use of IDACombo to computationally predict drug combination efficacy using single agent screening data.

**Validation of CTRPv2/GDSC drug combination predictions in NCI-ALMANAC.** We also sought to determine whether or not efficacy predictions made with IDACombo using monotherapy data from one dataset would be consistent with measured drug combination efficacies in a separate dataset, where the screening methodology and set of screened cancer cell lines differed. To this end, we used the Genomics of Drug Sensitivity in Cancer

**IDACombo predictions of clinical trial power closely agree with results for clinical trials in chemo-naïve patients, but not for trials in patients who had received previous treatment.** Following clinical trial and drug concentration selection, IDA-Combo was used to predict efficacies for the control and experimental treatments of each trial using the GDSC and CTRPv2 monotherapy cell line screening datasets. These two screens were chosen because they tested both a large number of compounds and cell lines. The large number of compounds

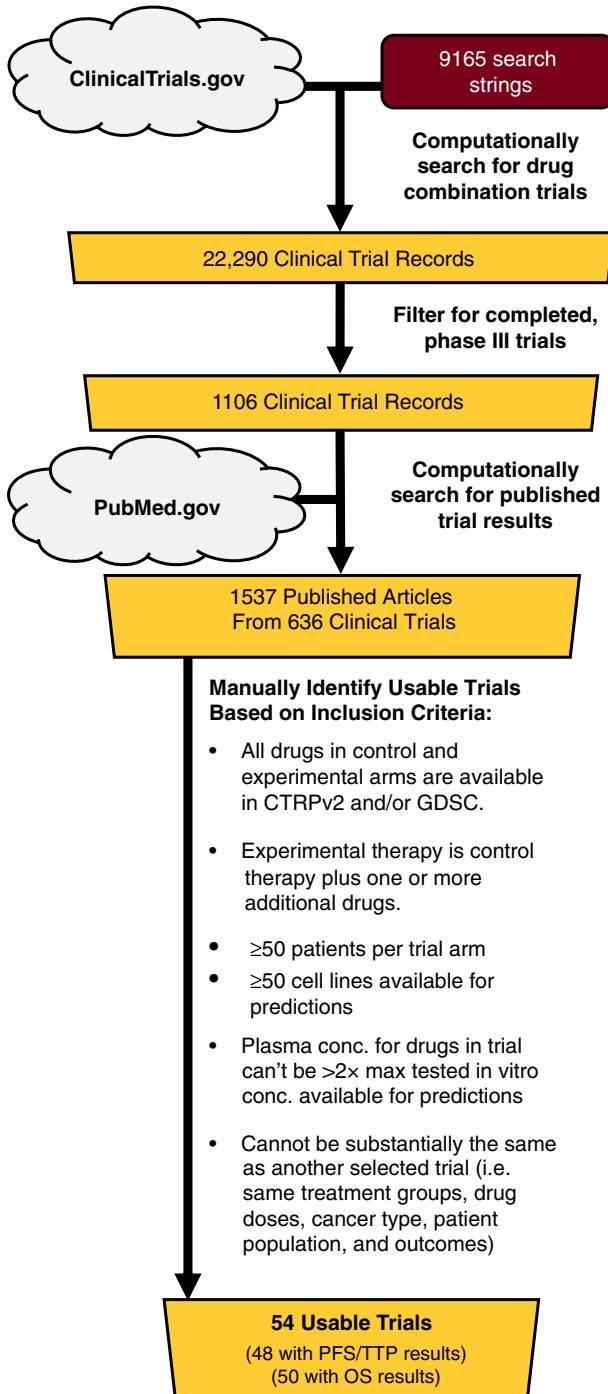

**Fig. 3 Trial selection pipeline for clinical validation.** Flowchart detailing how completed, phase III cancer clinical trials were selected for the clinical trial validation analysis. Searches of ClinicalTrials.gov and PubMed.gov were performed via web scraping (see "Methods" section) to identify published results for trials that may meet our inclusion criteria, and the identified clinical trial publications were then manually inspected to identify trials that met our study's inclusion criteria.

helped maximize the number of clinical trials we could make predictions for, and the large number of cell lines maximized the phenotypic diversity reflected in the predictions. The predicted efficacies were then used to calculate HRs between treatment groups in each trial. Encouragingly, despite making predictions for cancer-specific trials with a pan-cancer set of cell lines and the uncertainties in matching in vitro drug concentrations to

physiological drug concentrations, predicted HRs generated using cell line data were correlated with reported PFS/TTP HRs in clinical trials where patients had not received cancer drug treatment prior to trial entry (Fig. S5A) with a Pearson's $r = 0.60$ ($p = 0.0044$) and Spearman's rho $= 0.54$ ($p = 0.012$). It is notable, however, that, unlike in the in vitro validation analysis, predictions in this analysis tended to be overly optimistic, with predicted HRs generally being lower than reported PFS/TTP HRs.

To determine whether or not these predicted HRs are of sufficient quality to practically inform drug combination development, we used the predicted HRs to estimate the statistical power that each trial had to detect significant differences in PFS, TTP, or OS between the treatment groups and compared our predicted powers to whether or not each trial actually reported an improvement in these outcome metrics. These predicted powers are plotted in Fig. 4, with trials separated based on whether or not a statistical improvement in PFS/TTP or OS was reported in the published trial results. For our model to accurately predict trial results, we expect the higher the predicted power, the more likely we will see a trial to report a significant difference between treatment and control groups.

Using a standard 80% power cutoff to classify trials as likely or unlikely to detect a statistically significant improvement, our predicted powers for PFS/TTP correctly classified 84.6% of clinical trials in which patients had not received cancer drug treatment prior to trial entry (Fig. 4a), with >83% sensitivity and specificity. The analysis produced three false positives (indicated by gray arrows in Fig. 4a) and one false negative (indicated by an orange arrow in Fig. 4a), which are discussed in detail in the Supplemental Text. Briefly, three of the four misclassified trials were ovarian cancer trials, suggesting that our pan-cancer collection of cell lines may not perform well in predicting outcomes for this disease setting, and two of the three false-positives tested therapies that are currently in clinical use for those disease settings. Other possible explanations for the misclassification of these trials were also identified, including one of the trials being conducted exclusively in elderly adults with death from unknown causes/losing patients to follow up being considered as progression, and one of the tested drugs being tested with an improper solvent in CTRPv2—likely leading to it being inactivated and, therefore, not properly accounted for in IDACombo's predictions.

For OS powers in treatment-naïve trials (Fig. 4b), accuracy, sensitivity, and specificity were >90%, but it is difficult to confidently assess the suitability of IDACombo for predicting OS benefit, because we only identified three clinical trials in treatment-naïve patients which detected a statistically significant improvement in OS. Predicted HRs were associated with reported OS HRs for first-line therapy trials, but more weakly than with PFS/TTP HRs (Fig. S5B).

Unfortunately, the model performed much more poorly for clinical trials in patients who had undergone cancer drug treatment prior to entering the trial (Fig. 4c and d), reflecting the fact that no statistically significant association was detected between measured and predicted HRs in trials which involved previously treated patients (Fig. S5C and S5D). While the reasons for this poor performance in previously treated patients are not immediately clear, possible explanations are discussed later.

**The performance of predictions made with cancer type/subtype specific sets of cell lines is limited by the number of cell lines available for each cancer type**. Notably, the predictions in Fig. 4 were made using all of the available cell lines in CTRPv2 or GDSC, regardless of the cancer type being studied in each clinical trial. Analyses with cancer type/subtype-specific sets of cell lines

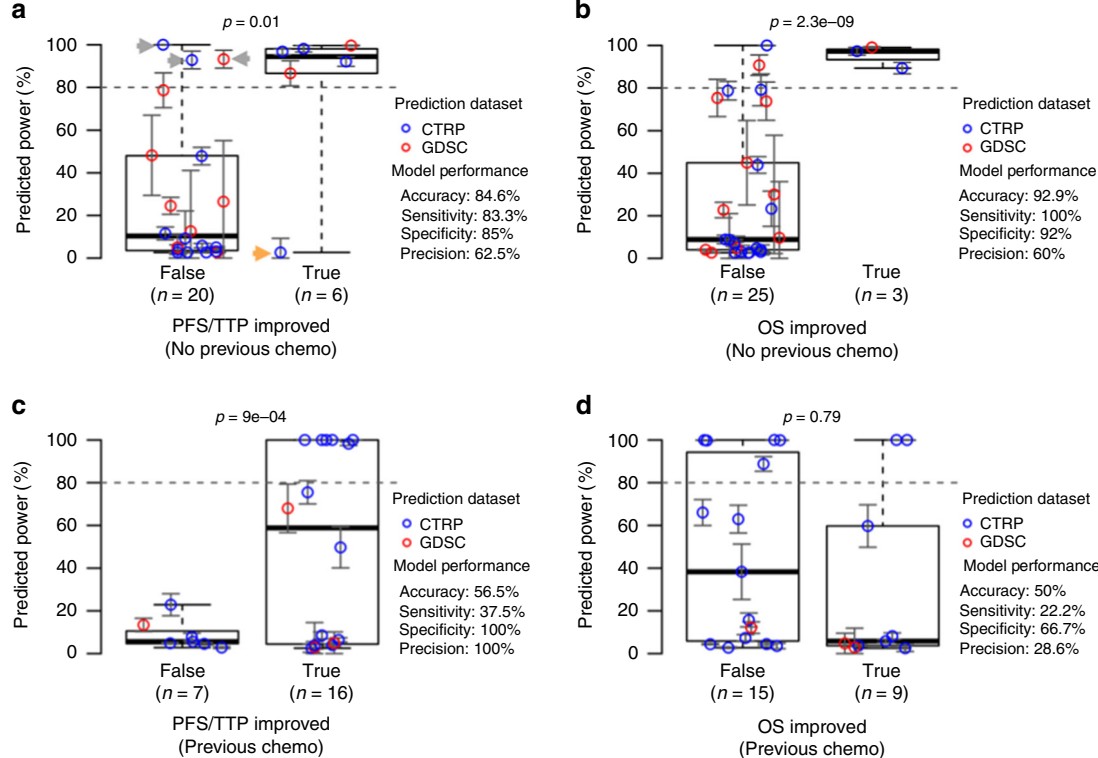

**Fig. 4 Clinical trial validation results show accurate efficacy predictions for trials in previously untreated patients but not for trials in previously treated patients.** IDACombo was used to make efficacy predictions for the control and experimental treatments of the clinical trials selected using the pipeline in Fig. 3. Hazard ratios were then calculated using these predictions, and study powers were calculated for each available comparison of a control therapy vs. an experimental therapy. These comparisons are separated based on whether or not the experimental arm statistically improved either PFS/TTP (panels **a** and **c**) or OS (panels **b** and **d**) in the published trial results. Predicted powers for each comparison are plotted on the y-axes, and an 80% power threshold (dashed line) is used to classify whether or not a comparison is expected to yield a statistically significant improvement. Comparisons are colored according to the dataset used to make predictions for the compared treatments. Panels **a** and **b** show results for trials in which patients had received no previous drug treatments. Panels **c** and **d** show results for trials in patients who had received previous treatment. Error bars for each plotted clinical trial power represent mean estimated power ± standard error (bounded between 0% and 100% power). Gray and orange arrows in Panel **a** indicate misclassified trials that are discussed in the text. P values were calculated using one-tailed t-tests. Blue circles indicate predictions made using the CTRP dataset, and red circles indicate predictions made using the GDSC dataset. Boxplots are plotted so that the lower and upper whiskers indicate the extreme lower and upper values, respectively, the box boundaries indicate the first and third quartiles, and the center line indicates the median. Source data are provided as a Source Data file.

were also performed and are presented in Fig. S6 and Supplementary Data 5. The results of these analyses are discussed in the Supplemental Text. Briefly, they suggest that predictions for targeted therapies are sensitive to the molecular features targeted by those therapies, but that predictions made using cancer-specific sets of cell lines generally perform less well than predictions made using all available cell lines due to the limited number of cell lines available for many cancer types. Thus, while the results of Fig. 4 indicate that predictions with pan-cancer sets of cell lines are largely sufficient for predicting the outcomes of cancer-specific clinical trials, IDACombo predictions may improve in the future if sufficiently large cancer-specific sets of cell lines can be used to generate cancer-specific predictions.

**Prediction performance is drug concentration dependent.** Beyond the selection of cell lines, we also investigated the importance of drug concentration selection for IDACombo predictions. The results of these analyses are discussed in the Supplemental Text. Briefly, they suggest that deviating from the identified clinical drug concentrations reduces model performance in the clinical trial validation (Fig. S7), and that IDA-Combo predictions lose accuracy when attempting to predict the efficacy of combinations with clinical drug concentrations >2× 

the maximum in vitro concentrations tested for those drugs (Fig. S8).

**Clinical predictions with Bliss Independence are less accurate than predictions with IDA.** To further validate our model, we compared our IDA-based results against predictions made using Bliss Independence[16]—one of the most established models for calculating the expected efficacy of a combination of non-interacting drugs (see "Methods" section). Bliss Independence-based predictions generally performed more poorly than IDA-based predictions (Fig. S9), suggesting that IDA is a more useful model for clinical drug combinations, at least for the trials in our dataset (see details in the Supplemental Text).

Overall, the clinical trial validation results suggest that IDACombo is capable of making highly accurate predictions of whether or not a clinical trial will detect statistically significant improvements in patient outcomes for drug combinations in previously untreated patients using only in vitro monotherapy information.

**Prospective pan-cancer IDACombo predictions reveal patterns based on drug mechanisms of action.** Given the encouraging validation results in both in vitro and clinical trial data, we chose

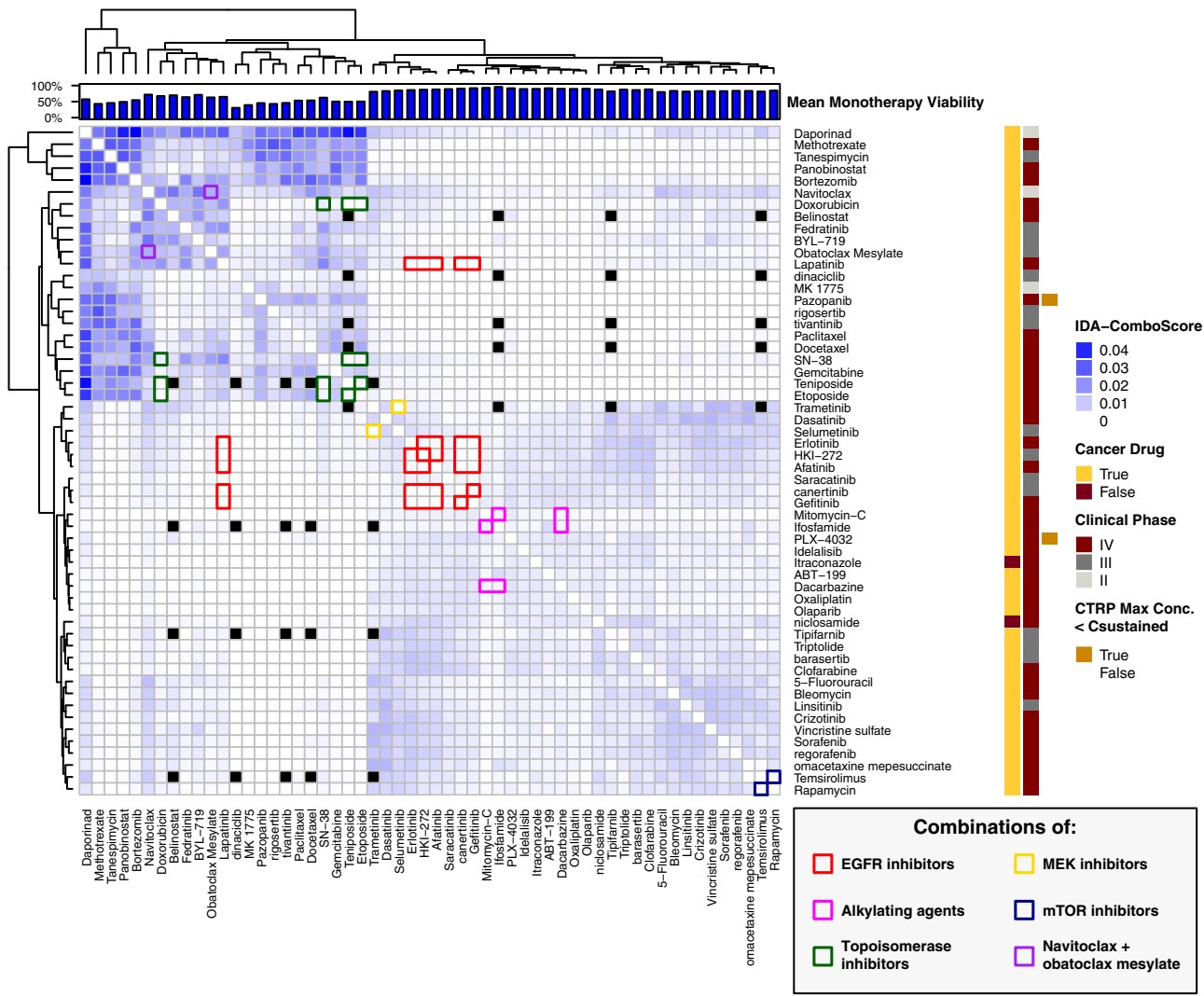

**Fig. 5 Top IDAcomboscore predictions for late-stage clinical drugs in CTRPv2.** IDAcomboscores were calculated for all two-drug combinations of late-stage clinical drugs in CTRPv2 using all available cell lines for each drug combination. Darker blue squares represent higher comboscores and, therefore, greater predicted drug combination efficacies relative to the constituent monotherapies. Black boxes represent missing values, where efficacies could not be predicted for a combination. The first bar, farthest left on the right side of the heatmap, indicates whether or not that drug is currently used for cancer treatment, the second bar indicates what stage of clinical trials that drug has reached, and the third bar indicates if the known Csustained concentration for the drug was higher than the maximum tested concentration in CTRPv2 such that predictions had to be made with a lower than clinical concentration. The barplot on top of the heatmap indicates the average viability achieved using that drug as a monotherapy (full bar indicates 100% viability). Drugs that were not predicted to combine well with any other drugs (i.e. with a comboscore < 0.004) were excluded from this plot to improve readability, but a full heatmap with all late-stage clinical drugs can be found in the "IDACombo Paper" project on OSF (see "Methods" section). Colored outlines are used to highlight certain combinations according to the "Combinations of" legend in the bottom right. Source data are provided as a Source Data file.

to create efficacy predictions for all possible two-drug combinations of clinically advanced drugs available in CTRPv2 or GDSC. However, the analysis of these results necessitated a different analysis strategy than was used for the clinical validation analysis. This was because power calculations were not convenient given the lack of knowledge about how many PFS/TTP or OS events would be observed in future trials of these combinations and the lack of knowledge about which control treatment each combination should be best compared to when calculating HRs. As a result, we developed the IDAcomboscore (see "Methods" section for details), which can be interpreted such that a higher IDA-comboscore indicates a more effective drug combination relative to the most effective single drug in the combination.

IDAcomboscores calculated using all available cell lines are plotted in Fig. 5 for combinations of CTRPv2 drugs which had at least one IDAcomboscore ≥ 0.004 (this cutoff was determined

based on heatmap cluster boundaries between drugs with higher and lower IDAcomboscores). Notably, the heatmap suggests that combinations of drugs which work via the same mechanisms of action are not expected to be more efficacious than the best monotherapy under IDA (see combinations of topoisomerase inhibitors, EGFR inhibitors, MEK inhibitors, mTOR inhibitors, or alkylating agents). This is not surprising, since IDA predicts that the best drug combinations will be comprised of drugs which act on completely separate populations of cells/patients, and drugs that have the same mechanism of action are likely to act on the same populations of cells/patients. An exception to this, however, can be found in the combination of navitoclax and obatoclax, which has a relatively high IDAcomboscore despite their both being classified as BCL inhibitors. The most likely explanation for this is that obatoclax has been found to have effects other than BCL inhibition. Indeed, it has been previously reported that

obatoclax can be highly effective in cell lines that are relatively resistant to navitoclax[17]. Unfortunately, an examination of combinations between drugs with different mechanisms of action (so-called "class effect") is more difficult than assessing combinations of drugs with the same mechanisms of action, because most mechanisms of action are only represented by a single drug in Fig. 5.

**The accuracy of prospective cancer-specific IDACombo predictions is currently limited by the number of available cell lines for each cancer type.** While the clinical validation analysis suggests that IDACombo predictions generated using pan-cancer sets of cell lines can be used effectively in predicting drug combination efficacy in many specific cancer types, we expect many researchers to be interested in making IDACombo predictions with cancer type-specific and subtype-specific sets of cell lines. Since many cancer types have few available cell lines, we sought to determine how many cell lines are necessary to create accurate predictions using IDACombo. The results of this analysis are discussed in the Supplemental Text, but, briefly, our findings suggest that prediction performance declines rapidly when predictions are made with <50 cell lines (Fig. S10A, Supplementary Data 6). Since most cancer types have <50 cell lines available (Fig. S10B, Supplementary Data 6), this suggests that cancer-specific predictions could be improved in the future by increasing the number of available cell lines for each cancer type.

**IDACombo predicts that navitoclax will efficaciously combine with taxanes in EGFR wild-type lung cancer.** To demonstrate how cancer-specific predictions can be used to identify novel efficacious drug combinations, we performed an example analysis in *EGFR* wild-type lung cancer (which is the cancer subtype with the most available cell lines in CTRPv2) aimed at identifying efficacious two-drug combinations with navitoclax, a BCL inhibitor currently in phase I/II clinical trials for lung cancer in various combinations. We began by identifying the highest predicted IDAcomboscores for navitoclax combinations in this cancer subtype (Fig. 6a). While it is possible to perform hypothesis testing using IDACombo to, for example, estimate the probability that a particular drug combination has an IDA-Comboscore ≥ a minimum desired IDAcomboscore, we chose to simply look at the top IDAcomboscores as there are several limitations to hypothesis testing with IDACombo which draw the robustness of such an approach into question (see the "Statistics" section in the "Methods" section for further discussion). The combination with the highest predicted efficacy was with daporinad (APO866), which is an NAMPT inhibitor that has yet to enter phase III trials after phase II trials failed to show significant efficacy as monotherapy in several cancer settings[18,19]. Strikingly, the second and fourth best combinations were both with taxanes (paclitaxel and docetaxel). Given the stalled clinical development of daporinad and the shared mechanism of action of paclitaxel and docetaxel, we decided to further investigate the combination of navitoclax with taxanes. By calculating mean viabilities for the combinations of navitoclax + taxane using a range of concentrations from 0 μM up to the achievable clinically sustained plasma concentration for each drug, we determined that the navitoclax + taxane combination is predicted to be superior to the best achievable monotherapy efficacy across a wide range of drug concentrations for both docetaxel and paclitaxel combinations (Fig. 6b and c). In fact, the analysis predicts that the maximal monotherapy efficacy can be achieved using combinations of the drugs at much lower doses (approximately one-third) than are required to achieve the same effect with any monotherapy. This is important, because it suggests that

combinations that are predicted to be efficacious by IDACombo may still be superior to monotherapy even if the clinical use of the combination requires lower doses to be used for each drug to limit toxicities.

Furthermore, other groups have tested the combination of navitoclax with taxanes in vitro, in vivo, and in phase I clinical trials. Their findings suggest that the combination shows superior efficacy to monotherapy regiments in pre-clinical tests and that it can be safely administered to patients[20–23]. Given these findings and our own predictions, we believe the combination of navitoclax with taxanes would be significantly more efficacious than either monotherapy alone in *EGFR* wild type lung cancer patients who have not received previous chemotherapy.

**IDACombo predicts that elesclomol can be efficaciously added to the combination of cisplatin and gemcitabine in EGFR wild type lung cancer.** To demonstrate how IDACombo can be used to identify candidate combinations of more than two drugs, we performed an analysis aimed at identifying drugs that could be added to the combination of cisplatin + gemcitabine to increase the treatment's efficacy in *EGFR* wild type lung cancer. Similar to the navitoclax analysis, IDAcomboscores were predicted for the addition of late-stage clinical drugs to the combination, with cisplatin + gemcitabine being treated as a monotherapy for the purposes of IDAcomboscore calculations. By far, the highest predicted IDAcomboscore was produced by the addition of elesclomol, an inducer of oxidative stress, to cisplatin + gemcitabine (Fig. S11A). Importantly, predicted IDAcomboscores remained high and predicted HRs remained low across a range of elesclomol concentrations (Fig. S11B and S11C), indicating that the combination is predicted to be superior to cisplatin/gemcitabine or elesclomol alone even if elesclomol must be used at low doses for toxicity reasons. We are unaware of any published studies testing the combination of cisplatin + gemcitabine + elesclomol. However, elesclomol has been demonstrated to selectively target cisplatin-resistant lung cancer cell lines[24], suggesting that it may target a phenotypically distinct population of tumor cells from that targeted by cisplatin. The high IDAcomboscore predicted for this combination supports this possibility and suggests that this combination should be further investigated for use in the first-line therapy of *EGFR* wild type lung cancer. Along with the navitoclax findings above, these findings demonstrate the ease and feasibility with which IDACombo predictions can be used to identify novel drug combination candidates for further development.

## Discussion
Our results demonstrate that IDACombo can be used with monotherapy cell line screening data to accurately predict drug combination efficacy both in vitro and in patients. While this does nothing to diminish the importance of continued efforts to understand and predict drug synergy/additivity, it does demonstrate that clinically meaningful predictions can be made using the simpler IDA hypothesis while methods which account for additivity and synergy continue to improve. In addition, our results are notable because they demonstrate that in vitro drug screening data can be used to generate clinically meaningful predictions for drug combination efficacies in patients despite the physiological and pharmacological differences between in vitro and in vivo systems. Furthermore, they suggest that many of these predictions can be made using pan-cancer sets of cell lines despite the wide range of genetic and phenotypic diversities observed among different cancer types.

There are, however, several limitations to this method. The first major limitation is an apparent unsuitability of cell-line-based IDACombo predictions for patients who have undergone

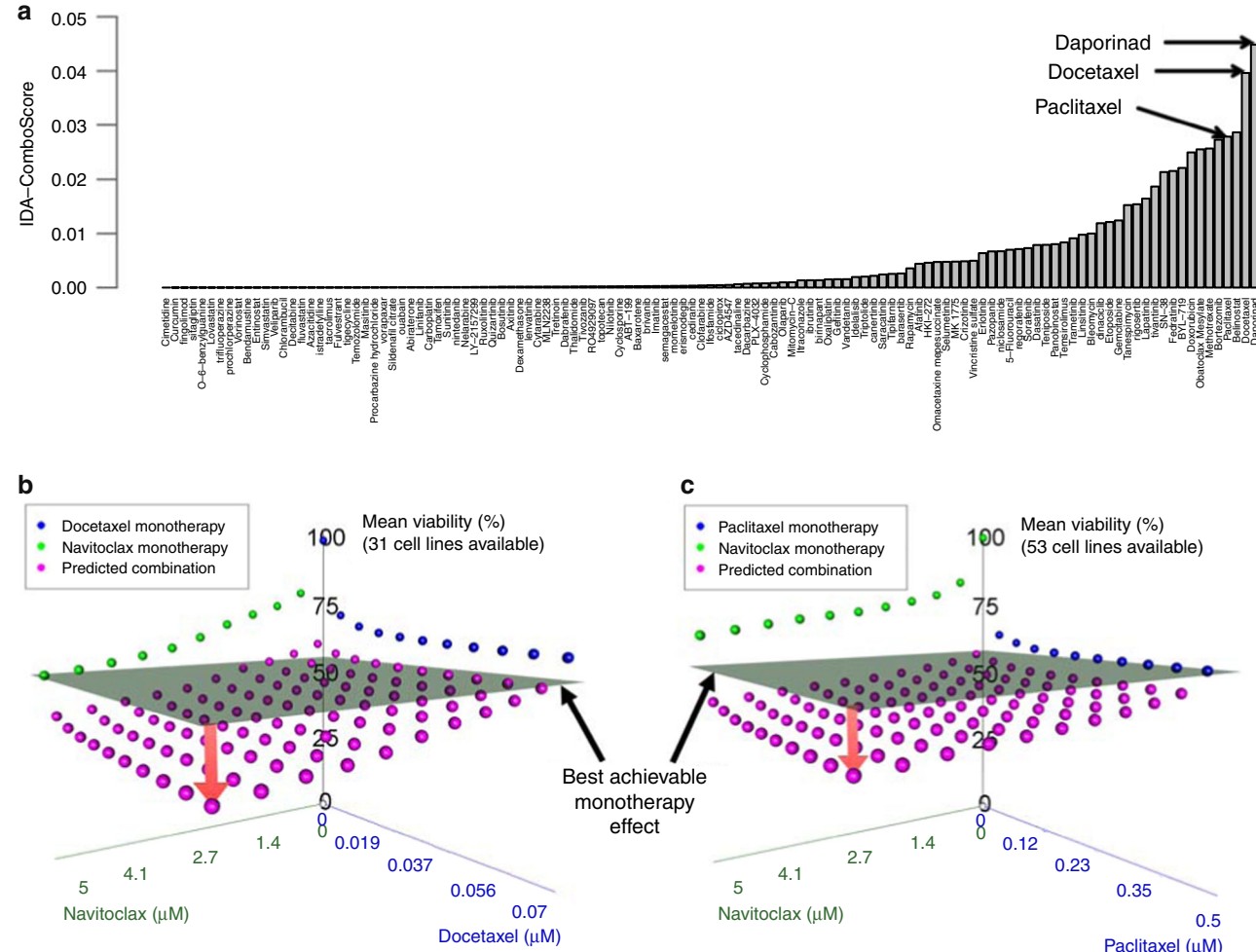

**Fig. 6 IDA-Combo predicts strong benefits for combinations of navitoclax and taxanes in *EGFR*-WT lung cancer. a** An ordered bar plot of the IDA-comboscores predicted using *EGFR*-WT lung cancer cell lines for combinations of navitoclax with other drugs that have reached late-stage clinical trials. Each bar represents a different combination of navitoclax with another drug. **b** and **c** 3-D plots of measured and predicted average cell viabilities at different concentrations of navitoclax and docetaxel **b** or paclitaxel **c**. The gray plane represents the lowest average viability achievable with monotherapy. The red arrow represents the difference between the best observed monotherapy effect and the best predicted combination effect, which suggests that the combination therapy will reduce tumor cell viability below what is achievable with monotherapy alone. Source data are provided as a Source Data file.

previous cancer drug treatment. One possible explanation for this is that the set of cell line models we used for model construction may be more representative of a population of treatment naïve tumors than it is of a population of tumors which have recently undergone a narrow set of treatments. It is well established that drug treatment can induce clonal selection in tumors in ways that alter the tumors' drug sensitivities[25]. While these altered sensitivities may be reflected in cell lines that were generated from the tumors of previously treated patients[26], it is likely that the cell lines in CTRPv2 and GDSC were derived under a diverse set of circumstances. This likely means that our set of cell lines contains greater phenotypic diversity at the population level than would be expected from a set of tumors which had recently undergone similar treatments and, therefore, similar selective pressures. Importantly, we do not believe this limitation negates the clinical value of our method. Of the trials systematically selected for our study, we identified more phase III cancer drug combination clinical trials in treatment-naïve patients than in previously treated patients, and more than half of the trials in treatment-naïve patients were published within the last decade. This suggests that there is still value in developing novel first-line therapies, meaning that drug combinations prioritized by IDACombo

may have immediate clinical value. A second limitation of this method is the inability of high-throughput cell line screens to measure the efficacy of immunotherapies—making it impossible for IDACombo to use currently available screens to predict the efficacy of drug combinations which include immunotherapies. It should be noted, however, that this does not mean that IDA-based predictions of drug combination efficacy are necessarily unsuitable for immunotherapies. Rather, this is currently a technical limitation that may be overcome by efforts to generate in vitro models suitable for screening immunotherapies[27]. These limitations, and potential strategies to overcome them, are discussed in more detail in the Supplemental Text.

Despite the limitations of this analysis, our results strongly support the notion that IDA is sufficient to explain the activity of many drug combinations used to treat cancer, and IDACombo provides a framework for translating monotherapy cell line data into clinically meaningful predictions of drug combination efficacy. Critically, while it is currently infeasible to experimentally test the vast number of possible cancer drug combinations, the algorithmic simplicity of IDACombo could allow researchers to computationally predict the efficacies of hundreds of millions of drug combinations in a matter of weeks to months. Moreover, as

new cancer cell line models are developed and cell line screens increase in size and quality, it should be possible to increase the accuracy and disease/subtype specificity of IDACombo's predictions well beyond the promising first-steps presented here. By using these predictions to systematically prioritize promising combinations for future experimental and clinical validation, this approach has the potential to significantly reduce the number of drug combinations which need to be tested—greatly reducing the cost and time of novel cancer drug combination discovery.

## Methods

### IDA and Bliss Independence drug combination efficacy predictions with IDACombo.

As shown in Fig. 1a, IDA predictions of drug combination efficacy are produced by predicting that the effect of a combination of two or more drugs on a cell line will be equal to the effect of the single most efficacious drug in the combination. The efficacy of the drug combination is then summarized by calculating the mean predicted efficacy across all cell lines being used in the analysis, and this average efficacy is used in downstream analyses. This equates to Eq. (1) below, where $\mu_{combo,IDA}$ is the mean IDA predicted efficacy of a combination of drugs $A$ to $Z$ in $n$ cell lines and where $EA_k$ and $EZ_k$ are the respective efficacies of drugs $A$ and $Z$ in cell line $k$.

$$\mu_{combo,IDA} = \frac{\sum_{k=1}^{n} \min(EA_k, \ldots, EZ_k)}{n}. \tag{1}$$

Note that this is well defined for any efficacy metric where a decrease in the efficacy metric indicates a decrease in the ability of a drug to kill cells (i.e. for a metric such as viability relative to untreated cells). If a decrease in the efficacy metric indicates an increase in the ability of a drug to kill cells (i.e. if the used metric is viability reduction, etc.), then the equation must be modified to Eq. (2).

$$\mu_{combo,IDA} = \frac{\sum_{k=1}^{n} \max(EA_k, \ldots, EZ_k)}{n}. \tag{2}$$

Bliss Independence predictions of drug combination efficacy are based on rearranged equations from Bliss[16], while assuming that the coefficient of association between drugs in a combination is equal to 0. This equates to Eq. (3) below, where $\mu_{combo,Bliss}$ is the mean Bliss Independence predicted efficacy of a combination of drugs $A$–$Z$ in $n$ cell lines and where $PA_k$ and $PZ_k$ are the respective probabilities of an individual cell surviving treatment with drugs $A$ and $Z$ in cell line $k$. These probabilities can be taken as the viabilities of cell line $k$ when treated with drugs $A$ or $Z$ relative to an untreated control.

$$\mu_{combo,Bliss} = \frac{\sum_{k=1}^{n} PA_k \times \ldots \times PZ_k}{n}. \tag{3}$$

Note that Bliss Independence is only defined for probabilities between 0 and 1, so any viabilities which fell below 0 or above 1 were rounded up to 0 or down to 1, respectively, for Bliss Independence calculations in our analysis.

### Percent viability.

Percent relative viability (shortened to "percent viability" throughout this paper) is simply the ratio of the viability of a treated cell line at a study's endpoint divided by the viability of an untreated control at the study's endpoint. As such, it can be interpreted such that 0% viability indicates complete cell death at a study's endpoint and 100% viability indicates identical viability to an untreated control at the study's endpoint. Notably, this means that percent viability is not able to differentiate between treatments that are cytotoxic and treatments that are cytostatic.

### Processing CTRPv2 and GDSC cell line drug screening data.

CTRPv2 and GDSC often use slightly different names for the same drugs and cell lines, so these identifiers were matched between the two datasets using the harmonized identifiers provided by Ling et al.[12]. The code used to do this is included in the "Harmonizing GDSC and CTRPv2" folder of the "IDACombo Paper" project on Open Science Framework (OSF, see "Data availability" statement).

Following identifier harmonization, four-parameter log-logistic dose–response curves were fit to the raw drug response data using the drc R package v3.0.1[28] and the code included in the "Reprocessing raw CTRPv2 and GDSC data" folder of the "IDACombo Paper" OSF project. This was done because the available sources of processed dose–response data for CTRPv2 and GDSC were generated using different algorithms between the two datasets. Recalculating the curves from the raw data allowed us to harmonize the analysis method for both datasets, and it allowed us to utilize information from all raw data points when estimating uncertainties in downstream analyses. Uncertainties in estimated viabilities were estimated using the drc R package predict.drc function and the sandwich R package v2.4.0[29,30] for calculating the variance–covariance matrix. Having fitted dose–response curves for each drug/cell-line pair in these datasets was necessary for this analysis because Csustained concentrations were often not tested directly in the datasets, so the curves were used to estimate these Csustained viabilities.

### NCI-ALMANAC analysis.

IDACombo was used to predict drug combination efficacies for the combinations included in NCI-ALMANAC using the mono-therapy data in NCI-ALMANAC. To avoid evaluating the accuracy of IDACombo for the same drug combination more than once, predictions were only made for each drug combination using the maximum tested monotherapy concentrations for each drug in the combination. Since Holbeck et al.[13] reported protocol differences between the different screening sites used to create NCI-ALMANAC, data for each monotherapy and drug combination was restricted to whichever site performed the most experiments for that monotherapy/combination. Furthermore, if multiple experiments were performed for the same treatment/cell line pair, the results of those experiments were averaged. The monotherapy-based drug combination efficacy predictions for each cell line were then averaged across all cell lines to produce a mean predicted efficacy for each drug combination, and the measured efficacies in NCI-ALMANAC were also averaged to produce a mean measured efficacy for each drug combination. These predicted and measured mean efficacies were then compared. All data and code used for this analysis is included in the "NCI-ALMANAC Analysis" folder of the "IDACombo Paper" project on OSF.

### AZ-S DREAM Challenge and O'Neil et al. 2016 Analysis.

As with the NCI-ALMANAC analysis, IDACombo was used to predict drug combination efficacies for the combinations included in the AZ-S DREAM Challenge dataset and the O'Neil et al. (2016) dataset using the monotherapy data in AZ-S DREAM and O'Neil et al., respectively. To avoid evaluating the accuracy of IDACombo for the same drug combination more than once, predictions were only made for each drug combination using the maximum tested monotherapy concentrations for each drug in the combination. The code used to perform these analyses has been included in the "AstraZeneca-Sanger DREAM Analysis" and "O'Neil 2016 Analysis" folders of the "IDACombo Paper" project on OSF. See the "Data availability" section for descriptions of where data was accessed for each of these datasets.

### Comparing IDACombo predictions made with CTRPv2 and GDSC to measured drug combination efficacies in NCI-ALMANAC.

Similar to the NCI-ALMANAC, AZ-S DREAM Challenge, and O'Neil et al. Analyses, monotherapy data from CTRPv2 and GDSC were used with IDACombo to predict mean combination viability for drug combinations in NCI-ALMANAC that consisted of drugs tested in CTRPv2 or GDSC, respectively. Note that predictions were only made for the maximum tested NCI-ALMANAC concentrations for each drug in the combination and that overlapping combinations were excluded if the concentration tested in NCI-ALMANAC exceeded the maximum tested concentration in CTRPv2 or GDSC, respectively, for any drug in the combination. All available cell lines in CTRPv2 or GDSC were used when making predictions for each drug combination. These CTRPv2 and GDSC predictions were then compared to the mean viabilities for the combinations that were experimentally measured in NCI-ALMANAC. Note that cell line overlap was intentionally not considered in this analysis, so CTRPv2/GDSC predictions were made with different (and much larger) sets of cell lines than were used to generate the measured NCI-ALMANAC efficacies. The data and scripts used in this analysis are included in the "NCI-ALMANAC Analysis" folder of the "IDACombo Paper" project on OSF.

### Identifying clinical trials for IDACombo clinical validation.

As outlined in Fig. 3, the rvest R package v0.3.2[31] was used to search ClinicalTrials.gov with 9165 search strings designed to identify trials that tested at least two of the drugs in CTRPv2 or GDSC. Search results were then compiled, resulting in the identification of 22,290 clinical trial records. These records were filtered to identify only completed, phase III clinical trials, resulting in 1106 clinical trial records. Web scraping with rvest was then performed again on ClinicalTrials.gov to search the records of each trial for listed publications associated with the trial. This resulted in the identification of 1537 publications associated with 636 clinical trials. Web scraping with rvest was then performed on PubMed.gov to collect the abstracts for each of these publications, which were then manually inspected to determine if the trial met the following inclusion criteria: 1. Completed, phase III clinical trial; 2. ≥50 patients per trial arm; 3. All cytotoxic drugs in control and test therapies are available in at least one of either CTRPv2 or GDSC; 4. ≥50 cell lines available for predictions of tested control and test therapies; 5. Test therapy is control therapy plus one or more additional drugs; 6. Clinically relevant drug concentrations for each drug in a trial are not >2× the tested drug concentrations in the dataset(s) necessary to predict that trial's efficacy (i.e. CTRPv2 and/or GDSC); and 7. Trial is not substantially the same as another selected trial (i.e. same treatment groups, doses, cancer type, patient population, and outcomes). These criteria were established prospectively with the exception of the criteria that clinically relevant drug concentrations for each drug in a trial must not be >2× the tested drug concentrations in the in vitro dataset(s) used to make predictions for that trial as this was not a problem we were expecting when the criteria were initially defined.

After trials were selected based on publication abstracts, the full articles were downloaded and reviewed for final selection and collection of trial information. This resulted in the identification of 54 clinical trials for use in the validation analysis—48 of which reported PFS/TTP results and 50 of which reported OS results. These trials tested 62 unique drug treatments (46 unique control vs. test treatment comparisons) involving 32 unique drugs. If it was discovered that a long-

term follow-up had been published for one of the selected trials, the most recent publication of the trial's results was used. The selected trials are listed in Supplementary Data 3. The data and code used in this selection process are included in the "Identifying Clinical Trials" folder of the "IDACombo Paper" project on OSF.

**Identification of clinically relevant drug concentrations**. In order to ensure that our drug combination efficacy predictions are clinically relevant, we surveyed the published literature to identify clinical plasma concentrations for all of the late-stage clinical drugs in CTRPv2 or GDSC. For drugs involved in the clinical trials identified for the clinical validation of IDACombo, we searched for plasma concentrations produced by the drug doses used in those trials. As such, multiple concentrations were identified for some drugs, each corresponding to a different dose of that drug. When multiple concentrations existed for a drug, the highest clinical concentration was used for the prospective analysis. All clinical concentrations as well as their corresponding citations are included in Supplementary Data 4, with concentrations for the clinical trial analysis in the first sheet of the table and concentrations for the prospective analysis in the second sheet.

Clinical concentrations were defined using published clinical trials which measured patient plasma concentrations over time after drug administration. Since many drugs that are administered via bolus IV exhibit extremely high plasma concentrations at the time of administration with a very rapid decrease in concentration immediately after administration, we decided that Cmax values were not appropriate for use in our model. As such, we opted to define our clinical concentration as the maximum plasma concentration achieved at least 6 h after drug administration, which we called Csustained. We chose 6 h because we observed that the exponential decline in plasma concentration for bolus IV drugs is typically finished by 6 h, though we also found that the idea of using 6 h plasma concentrations to define clinical drug activity is not unique to our study[32]. A graphical demonstration of how Csustained values were determined is included in Fig. S4.

**Estimating clinical trial powers with IDACombo**. Cell line viabilities and their associated uncertainties were estimated at Csustained drug concentrations using the fitted four-parameter log-logistic dose–response curves calculated from the raw CTRPv2 and GDSC data and the drc R package predict.drc function and the sandwich R package v2.4.0[29,30] for calculating the variance–covariance matrix. These monotherapy viabilities were then used to estimate mean viabilities for the control and test treatments using IDACombo either using IDA-based predictions or Bliss Independence-based predictions. To make a comparison between IDACombo predictions and clinical trial results possible, we chose to treat the remaining percent viability after drug treatment as an estimate of hazard for patients treated with that therapy, where 100% viability indicates a hazard of 1 (all cancer cells are alive relative to untreated) and 0% viability indicates a hazard of 0 (all cancer cells are dead). This allowed us to then estimate a HR for each control/test treatment comparison by dividing the mean test treatment viability by the mean control treatment viability.

It should be noted that it is an obvious simplification to assume that mean viability in cell lines is quantitatively linked to hazard for tumor progression or death in treated patients. The rationale behind this simplification is that a treatment which kills 100% of treated cancer cells in vitro may be hypothesized to kill 100% of treated tumor cells in vivo resulting in a hazard for progression or death of zero and that two treatments which kill the same proportion of treated cancer cells in vitro may be hypothesized to perform similarly in vivo, resulting in a HR of 1. It is less obvious, however, that mean viabilities between 0% and 100% can be hypothesized to have a direct linear relationship to clinical hazards. It is also notable that this approach cannot account for factors that influence patient hazard beyond the toxicity of a therapy to tumor cells—such as patient performance status or immune function. Despite these limitations, this approach was chosen because we believed that any comparison of IDACombo's predictions to clinical trial results must be aimed at estimating trial power, and doing so requires estimating a HR. We believe that directly dividing predicted mean viabilities of the test and control treatments is the fairest way to accomplish this task, because it provides no opportunity for arbitrary manipulation of the results as might be the case if response thresholds were used to classify cell lines as "responders" or "non-responders" for the purposes of hazard calculations.

Estimated HRs were used to estimate PFS/TTP/OS power for each trial using the powerSurvEpi R package v0.0.9[33] and the number of PFS/TTP/OS events observed in each trial. While a full description of the method used for power calculations is too detailed to include here, it can be found in the powerSurvEpi reference manual (https://cran.r-project.org/web/packages/powerSurvEpi/powerSurvEpi.pdf) under the "powerCT.default0" function. All clinical trial power predictions were performed using the data and scripts included in the "Clinical Trial Validation Analysis" folder of the "IDACombo Paper" project on OSF.

**Prospective analysis**. The prospective analysis was performed using all drugs in CTRPv2 and GDSC that have reached phase III or IV clinical trials, with selected phase 2 drugs included based on our lab's interest. For each selected drug, cell line viabilities were estimated using drug concentrations from 0 to Csustained and the

fitted four-parameter log-logistic dose–response curves calculated from the raw CTRPv2 and GDSC data. These monotherapy viabilities were then used to estimate mean viabilities for the control and test treatments using IDACombo using IDA-based predictions both with all available cell lines and with cancer-specific sets of cell lines. The predicted mean drug combination viabilities were then used to calculate HRs between the predicted drug combination efficacy and the best monotherapy efficacy ($HR_{C/Mbest}$) in the same way as was done for the clinical trial analysis. Since this HR would not allow for comparisons of drug combinations that did not share the same most effective monotherapy, we developed an IDA-comboscore metric which is calculated using Eq. (4), where Δvia is equal to the mean viability when cell lines are treated with the best monotherapy (i.e. monotherapy resulting in the lowest mean viability) minus the mean viability when cell lines are treated with the drug combination.

$$IDAcomboscore = \Delta_{via} - \Delta_{via} \times HR_{C/Mbest}. \quad (4)$$

The resulting metric is larger for drug combinations that are expected to be more efficacious, and it rewards drug combinations that maximally decrease the mean cell line viability relative to monotherapy while also having a low HR relative to the most effective monotherapy in the combination.

All data and code used to perform the prospective analysis is included in the "Prospective Analysis" folder of the "IDACombo Paper" project on OSF. Notably, this folder also includes a subfolder, "./Outputs/Cluster_Heatmaps/", with efficacy prediction plots and tables for predictions made with all cell lines and with 27 cancer type/subtype specific sets of cell lines in both CTRPv2 and GDSC. For plots and tables of all cell line predictions, combinations are only included if at least 50 cell lines were available for predicting the efficacy of that combination. For cancer type/subtype specific predictions, at least three cell lines were required for a combination to be plotted.

**Statistics**. Standard errors for IDACombo's predicted HRs, powers, and IDA-comboscores were estimated using a semi-parametric bootstrap. Briefly, this procedure was performed in the following steps:

1. Distributions of the calculated monotherapy viability values used in each analysis were simulated for each cell line by randomly sampling values from normal distributions with means equal to the calculated viability values and standard deviations equal to the standard errors estimated for each calculated viability using the fitted-dose–response curves described in the "Processing CTRPv2 and GDSC cell line drug screening data" section of the "Methods" section.
2. Simulated monotherapy viabilities were used to calculate estimated therapy efficacies for each cell line under the relevant model assumptions for each simulation.
3. Cell lines were randomly sampled with replacement for each simulation.
4. Mean therapy efficacies were calculated for each simulation.
5. HRs, powers, and IDAcomboscores were calculated for each set of simulated therapy efficacies, and the standard deviation of these simulated statistics were calculated and used to estimate standard errors, 95% confidence intervals, and relevant p-values.

10,000 simulations were performed for each control/experimental treatment in the clinical trial validation analysis, and 1000 simulations were performed for each drug combination in the prospective analysis. It should also be noted that, when sampling viabilities for two therapies which were to be compared and which shared one or more compounds, standard normal deviates were used to match sampled viabilities for those shared compounds between the two therapies. While simulating curve parameters may be a more robust approach than using standard normal deviates, particularly if the concentrations of the shared compounds differ significantly between the two therapies being compared, the standard normal deviate approach was chosen due to its being simpler to implement, faster to run, and more versatile. In particular, this approach allows IDACombo to estimate uncertainties when using data where only measured viabilities and their standard deviations are known, such that no curve fit parameters exist to be simulated (this is the case for datasets such as NCI-ALMANAC, where a number of compounds were tested at only three concentrations, making it impractical to fit dose–response curves).

Importantly, the standard errors estimated in this analysis only account for random errors in the measured viability values provided by the cell line drug screening datasets and in random sampling from the cell line population being used to make predictions. Systematic errors, such as might be caused by imprecise drug dilution or cell line counting, variation in phenotype between different aliquots of a cell line, or different protocols for performing cell line screens, are not modeled here. Likewise, no effort is made here to estimate uncertainties arising from predictions being made with cell line populations that do not adequately represent the patient population the predictions are to be applied to, as how to quantify how representative a collection of cell-line models is for a given set of cancer patients is an open question in the field. Additionally, no efforts are made to estimate the uncertainties introduced by uncertainty in the in vitro drug doses which most closely mimic the in vivo effect of each drug on patient tumors. As such, the uncertainties estimated for values calculated by IDACombo should be

considered as the lower limits of uncertainty, and should not be considered to be particularly robust for hypothesis testing.

To determine if a predicted IDAcomboscore is >0, the null probability that the IDAcomboscores is ≤0 was estimated by dividing the number of simulated IDAcomboscores that are ≤0 by the total number of simulated IDAcomboscores. Multiple testing correction was performed by using the Benjamini–Hochberg procedure[34] to estimate false discovery rates (FDRs), with FDRs ≤ 0.05 being considered statistically significant.

**Reporting summary**. Further information on research design is available in the Nature Research Reporting Summary linked to this article.

## Data availability

Source data are provided with this paper. With the exception of the AstraZeneca-Sanger Drug Combination Prediction DREAM Challenge dataset (controlled access, see below), all raw data necessary to run the analyses in this paper, along with detailed readme files to aid investigators in navigating and understanding each analysis and script in the project, have been uploaded in their entirety to Open Science Framework (OSF) and are stored in the "IDACombo Paper" project[35], which can be publicly accessed at https://osf. io/sym6h/.

The NCI-ALMANAC dataset was accessed by downloading the "ComboDrugGrowth_Nov2017.zip" folder from the following link on 5/17/2019: https:// wiki.nci.nih.gov/download/attachments/338237347/ComboDrugGrowth_Nov2017.zip? version=1&modificationDate=1510057275000&api=v2.

The AstraZeneca–Sanger Drug Combination Prediction DREAM Challenge dataset was accessed by downloading the "DREAM_OI_matrices_final.zip" [syn18468836] and "OI_combinations_synergy_scores_final.txt" [syn18435126] files from Synapse.org on 2/ 13/2020. Note that this data is not included in the OSF repository for this paper because it is controlled access and cannot be distributed with this manuscript.

The O'Neil et al., 2016 dataset was obtained by downloading the supplemental data files associated with O'Neil et al.[7] using links from the following webpage on 1/31/2020: https://mct.aacrjournals.org/content/15/6/1155.figures-only

Release 6.0 of the GDSC dataset was downloaded from the following website on 2/26/ 2018: ftp://ftp.sanger.ac.uk/pub4/cancerrxgene/releases/release-6.0

The CTRPv2 dataset was downloaded from the following weblink on 2/26/2018: ftp:// anonymous:guest@caftpd.nci.nih.gov/pub/OCG-DCC/CTD2/Broad/ CTRPv2.0_2015_ctd2_ExpandedDataset/CTRPv2.0_2015_ctd2_ExpandedDataset.zip. Source data are provided with this paper.

## Code availability

The code necessary to reproduce the analyses in this paper, along with detailed readme files to aid investigators in navigating and understanding each analysis and script in the project, have been uploaded in their entirety to Open Science Framework (OSF) and are stored in the "IDACombo Paper" project[35], which can be publicly accessed at https://osf. io/sym6h/.

Most analyses were performed using R v3.4.2[36] with Microsoft R Open v3.4.2[37] and RStudio v1.1.463[38]. Processing of the raw dose–response data from CTRPv2 and GDSC was performed using the Mesabi compute cluster at the Minnesota Supercomputing Institute (MSI) at the University of Minnesota (http://www.msi.umn.edu) and R v3.4.4. The IDACombo R package created for this analysis is available on GitHub at https:// github.com/Alexander-Ling/IDACombo/. Additional R packages used in the analysis are listed in Table S1 along with their citations and web-links.

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

## Acknowledgements

We thank Gary Oehlert and Emily Kurtz at the University of Minnesota Statistical Consulting Center and Dr. Hae Kyung Im at the University of Chicago for taking the time to provide feedback on our method of estimating uncertainties in the efficacy predictions produced by IDACombo. In particular, Gary made several key contributions to the design of these methods and to our understanding of their limitations. We also thank the Minnesota Supercomputing Institute (MSI, http://www.msi.umn.edu) at the University of Minnesota for providing resources that contributed to the research results reported within this paper. One of the datasets used for the analyses described in this manuscript was contributed by AstraZeneca and the Sanger Institute in collaboration with Sage Bionetworks-DREAM Challenge organizers. It was obtained as part of the AstraZeneca–Sanger Drug Combination Prediction DREAM Challenge through Synapse ID [syn4231880]. This study was supported by NIH/NCI Grants R01CA204856 and R01CA229618. R.S.H. also receives support from a research grant from the Avon Foundation for Women and an OACA Faculty Research Development grant.

## Author contributions

A.L.: Study design, analysis, and manuscript preparation. R.S.H: Study design and manuscript preparation.

## Competing interests

The authors declare no competing interests.
