## [Peer Review File · Nature Communications]

This manuscript has been previously reviewed at another journal that is not operating a transparent peer review scheme. This document only contains reviewer comments and rebuttal letters for versions considered at *Nature Communications*.

Reviewers' Comments:

Reviewer #1:

Remarks to the Author:

Comment 1: ((Cross-) validation on other cell line panels). These analyses substantially improve the quality of the manuscript. The performance drops significantly when validating predictions from one screen on another screen. The additional analyses and explanations provided by the authors provides the necessary context to these analyses. This point is now considered satisfactorily addressed.

Comment 6(1). Determining an objective cutoff for what is considered an interesting combination on the cell lines. With this comment I was aiming to suggest a way to separate "positives" (where a combination shows "significant" promise over a reference) from "negatives" (where a combination is indistinguishable from a reference) based on the cell line data only. There are 2 aspects here: 1) selecting promising combinations and 2) selecting these promising combinations based on *cell lines only*. Aspect 1 is accomplished by the authors by using the remaining viability of the combi as a proxy for the HR of the combination and then using the power of the clinical trial to define positives. My suggestion was to use the cell line data ONLY for the definition of positives. In this way it remains a pure prediction based on the cell line data and there is no information leakage from the test data (trials) to the data on which the predictions are made. In addition it also makes the method more versatile as there may not always be a trial to allow the selection of positives/negatives based on the power in the trial. Fortunately, Supplementary figure S5 provides much needed insight by comparing the point estimates of the predicted and observed HRs. This result shows more or less the same result as in Figure 4 but now in a more straightforward and transparent way - another important addition to the manuscript. As the authors rightly point out now only a single analysis (PFS, No previous chemo) shows significant Spearman and Pearson correlations. Inspection of the predicted HRs shows that in all cases a threshold of ~ 0.75 on the predicted HR more or less separates positives from negatives on the cell lines - so such a purely cell line based separation does not differ (for these cases) substantially from the power-based selection. I do not want to belabor this point any further and would request the following:

- 1) Add a sentence in which it is mentioned that the hypothesis test (such as the one proposed in the Statistics section) could be used to select positive cases in the absence of trial data to perform such a selection;
- 2) For each of the four cases (one per panel) depicted in Supplemental figure S5, list the reduction in average viability in the cell lines for the experimental treatment over the control for the top three predicted HRs and the bottom three predicted HRs to provide the reader with an intuition of the effect sizes in these cases.

Comment 6(2). Hypothesis testing and multiple testing correction. I appreciate the effort the authors took to update the simulations of the null model. I think these can be further refined, but as stated above, I do not want to belabor this point any further and I can appreciate that the method serves as a prioritization method. However, even in that case it is useful to know what the FDR is - if we obtain 100% FDR we are not going to even initiate expensive validation experiments, so pure prioritization is not always useful. With the additions as requested in Comment 6(1) I consider this point closed. It is up to the readership of the journal to further judge the method as it is now sufficiently transparent.

Comment 7. Hazard ratios to estimate statistical power. As stated above the addition of the scatter plots is very enlightening and provides the reader with a more complete view of the data. With the additional requests associated with Comment 6(1) this point is now considered closed.

REVIEWERS' COMMENTS

Reviewer #3 (Remarks to the Author):

Comment 1: ((Cross-) validation on other cell line panels). These analyses substantially improve the quality of the manuscript. The performance drops significantly when validating predictions from one screen on another screen. The additional analyses and explanations provided by the authors provides the necessary context to these analyses. This point is now considered satisfactorily addressed.

Comment 6(1). Determining an objective cutoff for what is considered an interesting combination on the cell lines. With this comment I was aiming to suggest a way to separate “positives” (where a combination shows “significant” promise over a reference) from “negatives” (where a combination is indistinguishable from a reference) based on the cell line data only. There are 2 aspects here: 1) selecting promising combinations and 2) selecting these promising combinations based on *cell lines only*. Aspect 1 is accomplished by the authors by using the remaining viability of the combi as a proxy for the HR of the combination and then using the power of the clinical trial to define positives. My suggestion was to use the cell line data ONLY for the definition of positives. In this way it remains a pure prediction based on the cell line data and there is no information leakage from the test data (trials) to the data on which the predictions are made. In addition it also makes the method more versatile as there may not always be a trial to allow the selection of positives/negatives based on the power in the trial. Fortunately, Supplementary figure S5 provides much needed insight by comparing the point estimates of the predicted and observed HRs. This result shows more or less the same result as in Figure 4 but now in a more straightforward and transparent way - another important addition to the manuscript. As the authors rightly point out now only a single analysis (PFS, No previous chemo) shows significant Spearman and Pearson correlations. Inspection of the predicted HRs shows that in all cases a threshold of ~ 0.75 on the predicted HR more or less separates positives from negatives on the cell lines - so such a purely cell line based separation does not differ (for these cases) substantially from the power-based selection. I do not want to belabor this point any further and would request the following:

- 1) Add a sentence in which it is mentioned that the hypothesis test (such as the one proposed in the Statistics section) could be used to select positive cases in the absence of trial data to perform such a selection;
- 2) For each of the four cases (one per panel) depicted in Supplemental figure S5, list the reduction in average viability in the cell lines for the experimental treatment over the control for the top three predicted HRs and the bottom three predicted HRs to provide the reader with an intuition of the effect sizes in these cases.

Comment 6(2). Hypothesis testing and multiple testing correction. I appreciate the effort the authors took to update the simulations of the null model. I think these can be further refined, but as stated above, I do not want to belabor this point any further and I can appreciate that the method serves as a prioritization method. However, even in that case it is useful to know what the FDR is - if we obtain 100% FDR we are not going to even initiate expensive validation experiments, so pure prioritization is not always useful. With the additions as requested in Comment 6(1) I consider this point closed. It is up to the readership of the journal to further judge the method as it is now sufficiently transparent.

Comment 7. Hazard ratios to estimate statistical power. As stated above the addition of the scatter plots is very enlightening and provides the reader with a more complete view of the data. With the additional requests associated with Comment 6(1) this point is now considered closed.

AUTHOR REPLY

We thank the reviewer for taking the time to go through our manuscript once again, and we are pleased that the large majority of the reviewer's concerns are now sufficiently addressed. We have modified Figure S5 as requested (see next page). We have also added a sentence mentioning the possibility of hypothesis testing with IDACombo in the section of the paper demonstrating an example prospective analysis to identify combinations with navitoclax in EGFR-WT lung cancer. This sentence is as follows:

While it is possible to perform hypothesis testing using IDACombo to, for example, estimate the probability that a particular drug combination has an IDAComboscore \geq a minimum desired IDAComboscore, we chose to simply look at the top IDAComboscores as there are several limitations to hypothesis testing with IDACombo which draw the robustness of such an approach into question (see the Statistics section in the Methods for further discussion).

We once again thank the reviewer for his/her time and for substantially improving the quality of this manuscript.

Figure S5. Predicted vs measured hazard ratios for clinical validation analysis. This figure shows how hazard ratios (HRs) predicted with IDACombo (x-axes) compare to HRs reported by the clinical trials selected for the clinical trial validation analysis (y-axes). Note that, while this figure includes largely the same set of trials used in Figure 4 in the main text, some of those trials are not included in this figure because they did not report HRs. Red points represent trials which did not report a HR that was statistically less than 1, while green points represent trials that did report a HR that was statistically less than 1. Circles represent trials where the power predicted by IDACombo for that trial was <80%, while squares represent trials where the predicted power was $\geq 80\%$. Pearson's r and Spearman's ρ are reported alongside two-sided p -values for whether or not the measured correlation is significantly different from 0. **A)** Measured PFS/TTP HRs vs predicted HR in clinical trials where patients had not received chemotherapy prior to trial entry. **B)** Measured OS HRs vs predicted HR in clinical trials where patients had not received chemotherapy prior to trial entry. **C)** Measured PFS/TTP HRs vs predicted HR in clinical trials where patients had received chemotherapy prior to trial entry. **D)** Measured OS HRs vs predicted HR in clinical trials where patients had received chemotherapy prior to trial entry. Note that further information for these trials and IDACombo's predictions for them is included in Data S4. The tables below each plot indicate the change in predicted mean viability for the experimental therapy vs. the control therapy for the three highest predicted HRs and the three lowest predicted HRs from each panel (negative values indicate experimental therapy has lower predicted viability than control therapy). Source data are provided with this paper.